# Comparative Peptidomics Analysis of Milk Fermented by *Lactobacillus helveticus*

**DOI:** 10.3390/foods11233885

**Published:** 2022-12-01

**Authors:** Shuman Gao, Yang Jiang, Xinyi Zhang, Shumao Cui, Xiaoming Liu, Jianxin Zhao, Hao Zhang, Wei Chen

**Affiliations:** 1State Key Laboratory of Food Science and Technology, Jiangnan University, Wuxi 214122, China; 2School of Food Science and Technology, Jiangnan University, Wuxi 214122, China; 3International Joint Research Laboratory for Pharmabiotics & Antibiotic Resistance, Jiangnan University, Wuxi 214122, China; 4Wuxi Translational Medicine Research Center and Jiangsu Translational Medicine Research Institute Wuxi Branch, Wuxi 214122, China; 5National Engineering Research Centre for Functional Food, Wuxi 214122, China

**Keywords:** *Lactobacillus helveticus*, fermented milk, comparative peptidomics, cell envelope proteinase, bioactive peptide

## Abstract

*Lactobacillus helveticus* is one of the commonly used starter cultures for manufacturing various fermented dairy products. However, only a few studies have explored the cleavage region preference of *L. helveticus* with different cell envelope proteinase (CEP) genes. In the present study, we profiled the peptide composition of milk samples fermented by three different *L. helveticus* strains by means of peptidomics to illustrate their different proteolysis patterns. The result revealed that the differences in peptide profiles of milk samples fermented by different *L. helveticus* strains were mainly a result of variations in the peptide patterns of the casein fractions, which were correlated with CEP genotypes. This was mainly reflected in the extensiveness of the hydrolysis region of αS1-casein and the degree of β-casein hydrolysis. Bioactive peptides were mostly derived from the hydrolysis region common to the three *L. helveticus* strains, and DQHXN-Q32M42 fermentation resulted in the highest diversity and abundance of bioactive peptides and a significant antihypertensive effect in spontaneous hypertension rats.

## 1. Introduction

*Lactobacillus helveticus* is one of the commonly used starter cultures in the manufacture of fermented dairy products, such as yogurt, Swiss-type cheeses, and Italian cheeses [1] and has considerable proteolytic capacity [2]. During proteolysis, proteins are usually degraded into oligopeptides (2–20 amino acids) by the cell envelope proteinase (CEP) of lactic acid bacteria (LAB), and are then transported via peptide transport systems into cells for further degradation into shorter peptides or amino acids under the synergistic effect of various intracellular peptidases [3]. Among LAB, *L. helveticus* possesses the most diverse distribution of CEP genes, with five paralogs (PrtH1, PrtH2, PrtH3, PrtH4, PrtH1 Variant) identified at present [4].

Previous studies have shown variations in the ability of different *L. helveticus* strains to hydrolyze milk proteins, especially casein, and the milk fermented by *L. helveticus* strains with different CEP genotype led to peptide heterogeneity [5]. Sadat-Mekmene et al. [6] found that the hydrolysis kinetics of αs1-casein was enhanced in the presence of *L. helveticus* strains with both PrtH1 and PrtH2. Skrzypczak et al. [7] showed that *L. helveticus* strains expressing three CEPs exhibit a greater degree of casein hydrolysis than those expressing one or two CEPs. On the other hand, previous research has also indicated that milk fermentation by *L. helveticus* notably increased the quantity and variety of bioactive peptides, such as angiotensin-I converting enzyme inhibitory (ACE-I) peptides [8], antioxidant peptides [9], anti-inflammatory peptides, and cognitive-improving peptides [10]. Among them, the lactotripeptides, Ile-Pro-Pro (IPP) and Val-Pro-Pro (VPP), and the lactodipeptide Tyr-Pro (YP), which are released from β- and κ-casein and cannot be obtained directly via gastrointestinal digestive enzymes [11,12], have been identified as ACE inhibitors and displayed the ability to reduce blood pressure in vivo [13,14,15,16]. However, few studies have profiled the peptide composition of fermented milk samples by different *L. helveticus* strains by means of peptidomics.

Recent advances in peptidomics provide us with a tool of characterization of the peptide profiles of fermented milks by various *L. helveticus* strains with different CEP genotypes, as well as the cleavage region preference of the caseins by *L. helveticus* strains. Thus, in the present study, the peptide profile of milk fermented by three *L. helveticus* strains with different CEPs, ATCC15009, DQHXN-Q32M42, and DYNDL36-6, was analyzed. Quantification of the IPP, VPP, and YP was also carried out, and the antihypertensive effect of milk fermented by *L. helveticus* was evaluated through spontaneously hypertensive rats (SHRs).

## 2. Materials and Methods

### 2.1. Lactobacillus helveticus Strains

The three *L. helveticus* strains used in this study were ATCC15009, DQHXN-Q32M42, and DYNDL36-6. DQHXN-Q32M42 and DYNDL36-6 were isolated from naturally fermented dairy products in China, and had been deposited in the Culture Collection of Food Microorganisms (CCFM) of Jiangnan University (Wuxi, China); ATCC15009 was a type strain isolated from Swiss Emmenthal cheese.

### 2.2. Analysis of CEP Genes

The blastn tool was used to analyze the CEP genes, with a BLAST identity cutoff of 98%. The sequences of CEP genes were obtained from NCBI, and the accession numbers were as follows: AAD50643 (PrtH1), ABI13574 (PrtH2), AER42337 (PrtH3), AER42338 (PrtH4), and ADX70200.1 (PrtH1-Variant).

### 2.3. Preparation of Fermented Milk Samples

The preparation method described by Wang et al. [17] was used with moderate modifications. All three strains were sub-cultured three times in MRS medium and then twice in sterile reconstituted skim milk (11% *w*/*w*) at 37 °C prior to experimental use. After two washes in Tris-HCl (pH 6.5), 2% cultures with an initial culture concentration of 1–5 × 10^8^ CFU/mL were inoculated with sterile reconstituted skim milk (11% *w*/*w*) and incubated for 48 h at 37 °C.

### 2.4. Preparation of Whey Fractions

The whey fraction was prepared as described previously [18] with adaptations. Briefly, trichloroacetic acid solution (TCA, 10% *w*/*v* in water) was added to fermented milk samples until the pH reached 4.6, and the supernatants were harvested after centrifugation at 10,000× *g* for 10 min at 4 °C. Then filtration with a 0.45 µm nylon syringe filter and ultrafiltration using a membrane with a cut-off value of 10 kDa were performed. The whey fractions were desalted using a C_18_ SPE column, collected, and dried by centrifugal evaporation at 40 °C for 2 h.

### 2.5. Identification of Peptide Sequences by UPLC-ESI-MS/MS

A method previously described [19] was followed with minor modification. Samples were analyzed using an EASY nLC 1200 system equipped with an Acclaim PepMapTM RSLC (50 μm × 15 cm, 2 μm, 100 Å). Mobile phase A was an aqueous solution containing 0.1% (*v*/*v*) formic acid and 2% (*v*/*v*) acetonitrile, while mobile phase B was an aqueous solution containing 0.1% (*v*/*v*) formic acid and 90% (*v*/*v*) acetonitrile.

The gradient elution procedure was as follows: 6–20% B, 0–40 min; 20–32% B, 40–52 min; 32–80% B, 52–56 min. The flow rate was set to 0.2 μL/min. The eluted peptides were directly ionized in a mass spectrometer (Thermo Scientific, Waltham, MA, USA) at an ion source voltage of 2.3 kV and subjected to a fully continuous MS scan followed by MS/MS scans three times. MS analysis was performed using the positive ion and data dependence analysis mode. The MS spectra were collected from 150–2000 M/Z with a resolution of 60,000. The MS/MS data acquisition range was 180–2000 M/Z, and the scanning time was 0.1 s.

Using the protein of *Bos taurus* (signal peptide excluded) from the Uniprot database (available online: https://www.uniprot.org/ (accessed on 5 January 2021)) as the comparison data, the original data were compared and retrieved using MaxQuant (1.5.2.8; Max-Planck Institute for Biochemistry, Martinsried, Germany). A non-specific enzyme cleavage mode was used, and the parameter settings for the database search were used as follows: mass tolerance, 0.02 Da; de novo synthesis tolerance,10 ppm; MS/MS deisotope tolerance, 7 ppm; false discovery rate threshold for protein and peptide matching, 1%. Each sample had three replicates; when the same peptide appeared twice, it was considered to be present in the sample. Subsequently, the peptides detected in each sample were uploaded to the Milk Bioactive Peptide Database (MBPDB) (http://mbpdb.nws.oregonstate.edu/, accessed on 4 June 2022) for functional searches [20].

### 2.6. Peptide Quantification

The IPP, VPP, and YP contents were determined by triple quadrupole linear ion trap liquid chromatography-mass spectrometry (AB Sciex, Framingham, MA, USA), and a suitable targeted mass spectrometry multi-reaction monitoring method was established. The standard samples of these three peptides were diluted into concentration gradients using 50% acetonitrile aqueous solution, and the contents of each peptide in the samples were calculated by external standard calibration.

### 2.7. Animals and Experimental Design

Sixteen-week-old male SHRs were purchased from the Beijing Vital River Laboratory Animal Technology Co., Ltd. (Beijing, China). All animal procedures were approved by the Ethics Committee of the Jiangnan University, China (JN. No 20210630W0661216 [245]). The rats were housed in an animal facility under specific pathogen-free conditions at the Laboratory Animal Center of the Department of Food Science and Technology, Jiangnan University, Wuxi, China. The rats were kept under the following conditions: 22 ± 1 °C, 55 ± 10% humidity, and a daily 12 h light/dark cycle.

A single intragastric administration experiment was started after one-week acclimatization. Rats were assigned to the vehicle, positive control, and fermented milk sample-treated groups (*n* = 6 for each group). Fermented milk sample-treated groups were administered a gavage dose of 10 mL/kg bodyweight, the given vehicle group was gavaged with 0.9% saline once daily with an equal volume, and the positive control group was gavaged with 100 mg/kg bodyweight captopril. All animals were provided ad libitum access to food and water. Blood pressure was measured before and 3 h after the gavage administration. After blood pressure readings stabilized, 10–12 additional consecutive readings were averaged.

### 2.8. Statistical Analysis

Log10 scale transformation was applied to carry out the statistical analysis of peptidomics, and principal component analysis (PCA) and partial least squares discriminant analysis (PLSDA) were performed to characterize the differences in peptidomic profiles in MetaboAnalyst 5.0 (available online: https://www.metaboanalyst.ca (accessed on 12 July 2022)) [21]. The heat maps of the peptidomics data were constructed using Peptigram, a web-based application for peptidomics data visualization [22], and the heatmap of bioactive peptides was drawn using TBtools [23]. Bar plots were constructed using GraphPad Prism 8.0 (GraphPad Software, San Diego, CA, USA). The mean ± standard deviation (SD) was used for the presentation of data, and one-way ANOVA with Tukey’s multiple comparison was carried out. Statistical significance was set at *p* < 0.05.

## 3. Results and Discussion

### 3.1. CEP Genes Distribution in Strains

The distribution of CEP genes in the selected strains is listed in Table 1. Three *L. helveticus* strains belong to different cell protease genotypes. More specifically, ATCC15009 possesses the *prtH3* gene, DYNDL36-6 possesses *prtH2* and *prtH4* genes, and DQHXN-Q32M42 possesses three genes (*prtH2*, *prtH3*, and *prtH4*), with the genotypes of *prtH3* and *prtH4* being rarely reported according to the literature [4,18].

### 3.2. Peptide Profile Analysis

The PCA across the peptides of skim milk and three fermented milk samples is shown in Figure 1a. Clear separation among the four samples was observed. The result indicated that all three *L. helveticus* strains significantly altered the peptide composition of the fermented milk samples, with variation observed among them. PLSDA revealed the differential peptides among the three fermented samples, with the top 20 shown in Figure 1b, ranging from 9 to 24 amino acids in length. DQHXN-Q32M42 possessed all 20 peptides, and 19 peptides were recorded in the ATCC15009 sample with αS2-casein f(149,162) missing, whereas DYNDL36-6 only possessed 2 out of 20 differential peptides, αS2-casein f(149,162) and αS2-casein f(100,115).

Figure 2a shows great variation in the ability of different *L. helveticus* strains to generate peptides, with the highest number of peptides (450) present in DQHXN-Q32M42 and only 181 in DYNDL36-6. These peptides mostly originated from β-casein (32%–51%), followed by αS1-casein (14%–30%), κ-casein (13%–15%), and αS2-casein (6%–10%) (Figure 2b), which is in line with the results reported by Fan et al. [24]. However, considering the fact that αS1-casein, β-casein, αS2-casein, and κ-casein (κ-CN) account for 30%, 28%, 8%, and 10% of caseins, respectively [25], β-casein seems to be preferred by the proteases of *L. helveticus* strains. In total, 65 common peptides were identified among the three samples, with 80% from casein, which was consistent with the percentage of casein among all the bovine proteins. Regarding the unique peptides, DQHXN-Q32M42 had the largest number with 262 peptides, followed by ATCC15009 (152) and DYNDL36-6 (76). The lengths of the peptides varied among the samples, with DQHXN-Q32M42 normally distributed around peptides with chain lengths of 10, and ATCC15009 and DYNDL36-6 around 14–16 (Figure 2c). The results also indicated the presence of a large number of tripeptides in the three fermented samples.

### 3.3. Correlation between CEP Genotypes and αS1- and β-Casein Cleavage Patterns

Figure 3 shows the proteolysis patterns of the casein fractions with the detected peptides mapped on the αS1- and β-casein sequences. αS1-casein was divided into eight regions, with regions I, IV, and VIII covering the main common sequences (Figure 3a; see Appendix A for specific details of the regions). There were significant differences in the sequence coverage for αS1-casein among the three fermented samples (55% in ATCC15009, 73% in DQHXN-Q32M42, and 37% in DYNDL36-6). In general, αS1-casein was mainly cleaved in regions I, IV, VI, and VIII by three strains, including the N- and C-termini. Sadat-Mekmene et al. [6] showed similar results and proposed that the lack of secondary structure at the N- and C-terminal ends of αS1-casein led to the susceptibility to hydrolysis in these regions. On the other hand, the micellar forms of casein would also influence the accessibility of different regions of caseins to the CEP of the strains.

Overall, DQHXN-Q32M42, the *L. helveticus* strain with *prtH2–3–4*, can cleave more regions of αS1-casein than the strain with only *prtH3* and that with *prtH2–4*. The common hydrolytic sections of II and V in αS1-casein were observed for ATCC15009 and DQHXN-Q32M42, which might be related to the possession of *prtH3* in these two strains. On the other hand, the regions of VI (131–142) were shared by DQHXN-Q32M42 and DYNDL36-6, which both possessed *prtH2* and *prtH4*. Further research with more strains of various CEP genotypes is necessary to interpret the pattern of cleavage regions.

Regarding β-casein, the peptides from three *L. helveticus* strains cover most of the sequence of β-casein, except for the N-terminal region (signal peptide excluded) (Figure 3b). All three *L. helveticus* strains were able to cleave over 70% of the β-casein (84% in ATCC15009, 75% in DQHXN-Q32M42, and 73% in DYNDL36-6). These findings are in agreement with the results of Begunova et al. [26]. Notably, β-casein seemed to be more accessible to cleavage and subsequently more hydrolyzed than the αS1-casein, probably because this molecule is considered to be an intrinsically unstructured protein [27]. The peptides from regions III and V of β-casein were shared by the three samples, with I (3–5, 22–27) being the unique section of ATCC15009. Section II is the common section for ATCC15009 and DYNDL36-6, and section IV is the common section for ATCC15009 and DQHXN-Q32M42.

However, as shown in Figure 2b, the number of peptides (ATCC15009: 176, DQHXN-Q32M42: 185, and DYNDL36-6: 58) derived from β-casein differed significantly. Further analysis indicated that DQHXN-Q32M42, the strain with *prtH2–3–4*, generated more widely distributed peptides than ATCC15009 (*prtH3*) and DYNDL36-6 (*prtH2–4*). For example, Figure 4 shows the distribution of various peptides from β-casein (142–166) among the three fermented milk samples. It can be observed that DQHXN-Q32M42 had more diverse cleavage sites than the other two strains, which was in agreement with the fact that the peptide length of DQHXN-Q32M42 was normally distributed around 10, while those of ATCC15009 and S12 were 14 and 16, respectively. Thus, differences in CEP genes affected the degree of β-casein hydrolysis, which was reflected by the length and number of peptide segments, but not the preference of regions of hydrolysis.

### 3.4. Bioactive Peptide and Function Confirmation

A MBPDB search was performed for the identification of known bioactive sequences, and the results are presented in Appendix A. Forty-five bioactive peptides with fourteen functions were identified, mainly including ACE inhibition activity, antioxidant activity, and antibacterial activity. DQHXN-Q32M42 fermentation resulted in the highest diversity and abundance of bioactive peptides, particularly of small bioactive peptides (fewer than 16 amino acid residues), followed by ATCC15009 and DYNDL36-6. These bioactive peptides were mainly from the common sections of αS1-casein (I, II, IV, and VIII) and β-casein (III and V), which might be because these regions can be hydrolyzed extensively (not only by *L. helveticus*) and have been widely studied. These findings were at odds with the findings of Skrzypczak et al. [28], who revealed that κ-casein proved to be the main source of short peptides released by bacterial enzymes. Because the presence of ACE-I peptides exceeded half of the number of bioactive peptides in the samples, these peptides were selected for further analysis. The distribution of the ACE-I peptides among three samples fermented by *L. helveticus* strains is shown in Figure 5a. Compared with the other two fermented milk samples, the sample fermented by DQHXN-Q32M42 had a higher diversity and abundance of ACE-I peptides. Even though all the strains could hydrolyze proteins to generate the peptide “LLYQEPVLGPVRGPFPIIV”, “EPVLGPVRGPFP” was only present in the samples of DQHXN-Q32M42 and ATCC15009, with “YQEPVLGPVR” and “PFP” only in the fermented milk of DQHXN-Q32M42. In short, these results suggested that the strain with *prtH2–3–4* had a stronger casein hydrolysis capability and was subsequently more prone to generate more and smaller bioactive peptides, which was further supported by the quantitative analysis of typical ACE-I peptides such as IPP and VPP. Quantitative results showed that the DQHXN-Q32M42 sample had the highest IPP and VPP contents among the samples, at 115.269 ± 13.058 and 13.472 ± 1.554 mg/L, respectively (Figure 5b). The milk fermented by DQHXN-Q32M42 had a significant antihypertensive effect (−11.62 ± 3.33 mmHg) in SHR (Figure 5c). Therefore, our study revealed that the existence of different CEPs in *L. helveticus* results in different peptide profiles of casein and ultimately impacts bioactivity, and that the strain with *prtH2–3–4* exhibited greater biological potential among the three strains. Further study with more *L. helveticus* strains is required to validate the mechanism of cleavage patterns of various CEP genotypes and the biological potential of milk fermented by *L. helveticus*.

## 4. Conclusions

Comparative peptidomics analysis revealed different peptide profiles in milk fermented by different *L. helveticus* strains. These differences were mainly attributed to variations in the peptide profiles of the casein fractions. The αS1- and β-casein hydrolysis patterns were correlated with the CEP genotypes of *L. helveticus* strains. Among the three *L. helveticus* strains, DQHXN-Q32M42 with *prtH2–3–4*, which possessed two unique hydrolysis regions III (50–63) and VI (143–157) of αS1-casein and generated smaller peptides from β-casein, demonstrated the highest casein hydrolysis activities. Bioactive peptides were mostly derived from the hydrolysis region common to the three *L. helveticus* strains, and DQHXN-Q32M42 fermentation resulted in the highest diversity and abundance of bioactive peptides and a significant antihypertensive effect in SHRs.

## Figures and Tables

**Figure 1 foods-11-03885-f001:**
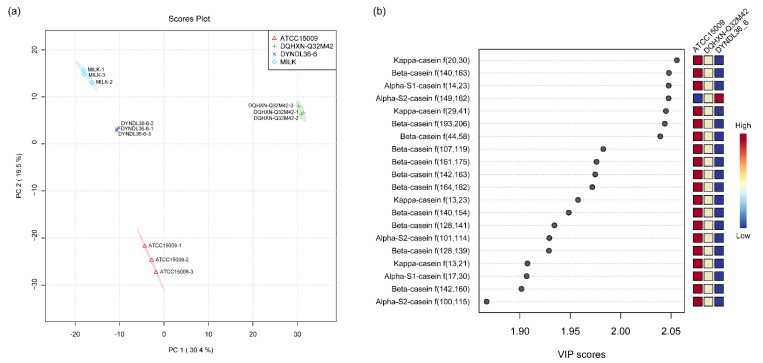
(**a**) Principal component analysis for peptidomics of fermented milks by *Lactobacillus helveticus*. (**b**) Peptides with the top 20 variable importance in projection (VIP) scores in the fermented milk of *L. helveticus* according to partial least-squares discriminant analysis.

**Figure 2 foods-11-03885-f002:**
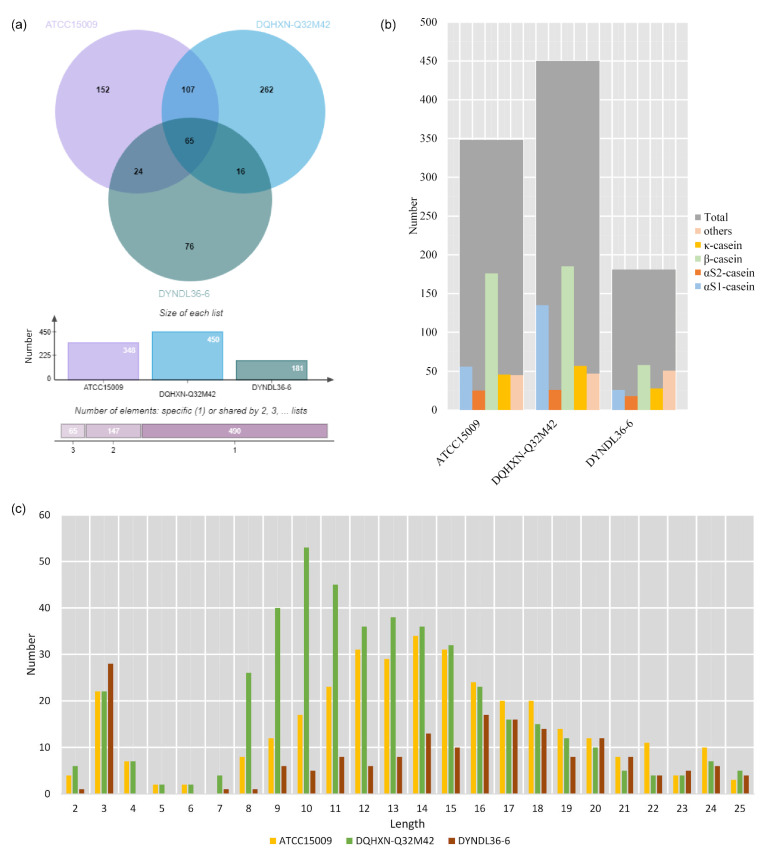
Set analysis of peptidomics data. (**a**) Venn diagram of the peptidomics of fermented milks by *Lactobacillus helveticus*. The top figure is a classical Venn diagram, the middle figure shows the number of peptides in every sample, and the bottom figure lists the numbers of common peptides and unique peptide. (**b**) Peptide source protein distribution map of the identified peptides. (**c**) Peptide length distribution map.

**Figure 3 foods-11-03885-f003:**
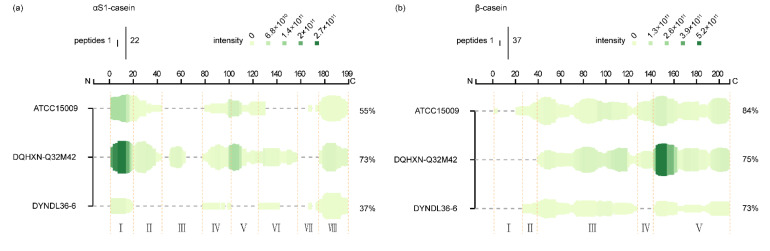
Heat map of (**a**) αS1-casein and (**b**) β-casein. The height of the green bars and the intensity of the green color are proportional to the count of peptides and the sum of the peptide peak intensities overlapping this position, respectively. The count of peptides overlapping a residue and the sum of their intensities is displayed at the top of the figure.

**Figure 4 foods-11-03885-f004:**
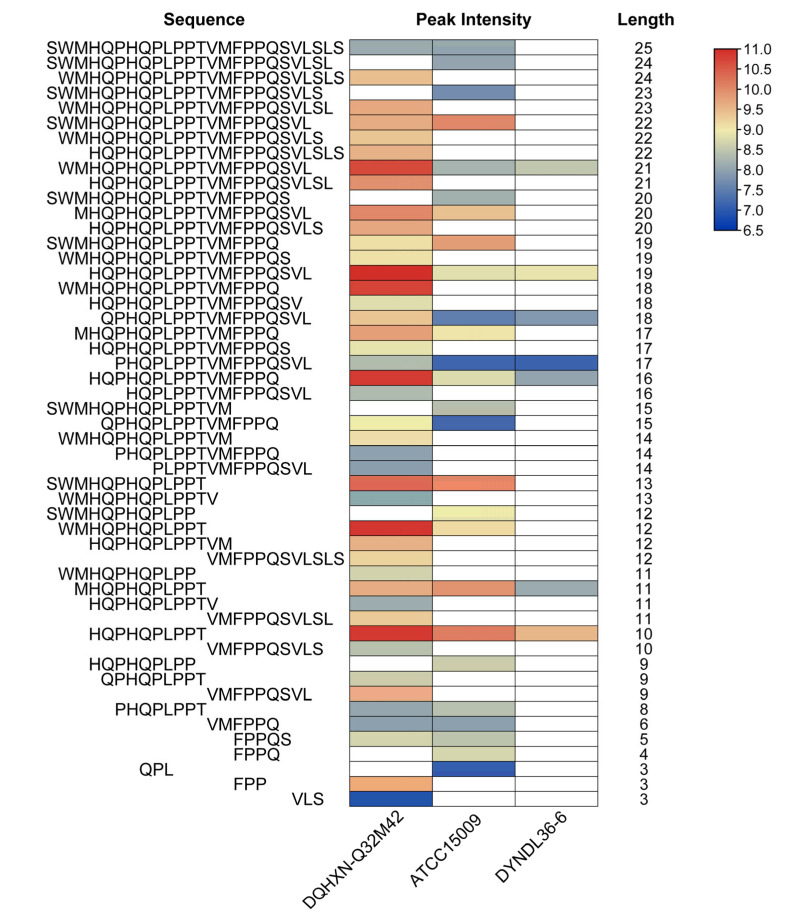
Heat map of peptides derived from β-casein (142–166).

**Figure 5 foods-11-03885-f005:**
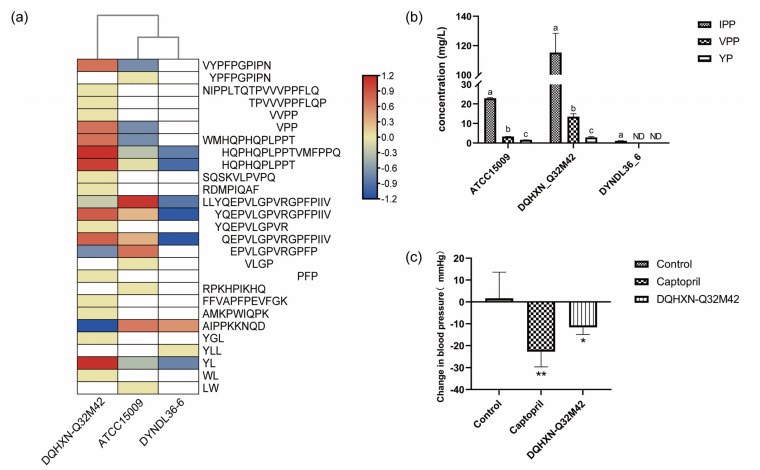
Distribution of ACE inhibitory peptides in *Lactobacillus helveticus*-fermented milk and their blood pressure regulation ability. (**a**) ACE-I peptides identified in fermented milk samples of *L. helveticus*. (**b**) Concentrations of IPP, VPP, and YP in the milks fermented by *L. helveticus*. Significant differences (One-way ANOVA with Tukey’s multiple comparison, *p* < 0.05) between groups are indicated with different letters (a, b and c) above the bars. ND means not detected. (**c**) Effects of single gavage of milks fermented by *L. helveticus* for blood pressure in early SHR. One-way ANOVA with Tukey’s multiple comparison; * *p* <0.05, ** *p* <0.01.

**Table 1 foods-11-03885-t001:** Distribution of CEP genes in experimental strains.

Strains	ATCC15009	DYNDL36_6	DQHXN-Q32M42
*PrtH1*	−	−	−
** *PrtH2* **	**−**	**+**	**+**
** *PrtH3* **	**+**	**−**	**+**
** *PrtH4* **	**−**	**+**	**+**
*PrtH1 Variant*	−	−	−

“+” means that the gene has been identified; “−” means that the gene has not been identified. Genes listed with bold font indicate that the genes were differentially expressed in strains.

## Data Availability

The datasets generated for this study are available on request to the corresponding author.

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
