# Peer review of "Comparative Peptidomics Analysis of Milk Fermented by Lactobacillus helveticus"

_foods, 2022, doi:10.3390/foods11233885_

Round 1

Reviewer 1 Report

Line 1: Change the entire document to MDPI Foods template (Check author guidelines: https://www.mdpi.com/journal/foods/instructions)                      

Line 20: The abstract should be a total of about 200 words maximum. Rewrite to reduce the word count.

Line 63: Needs further clarification on what exactly causes these changes

Line 55-65: The authors discuss well in this section, but there is no proper coherence, consequently, it should be rewritten. I ask authors to organize information. This version may make the reader confused. The final target should be clearly explained in the introduction. Could also explain why profiling the peptide composition of milk fermented by different strains are essential

Line 66-73: Very vague, can be rewritten to effectively give more context, include all latest publications

Line 77: Please clearly present the hypothesis and contribution of your study in the Introduction section.

Line 93: Briefly explain what the modifications were

Line 85: Was any other time considered other than 48 hrs  

Line 154: Experimental design is not clear. Was it a randomized approach to include all necessary experimental approaches in the text? Was it done in replications?

Line 172-176: Further discussion is required

Line 215: Further discussion is required

Line 184: interesting; please provide further discussion

Line 193: what does that indicate

Line 219: Typically, in MDPI Foods Results and Discussion are written together; consider combining  

Line 273: Being very wishful, please provide further clarifications from an industry standpoint

Line 287: Include a comment indicating the potential use of the proposed use at the industrial level.

*The figures and tables are missing in the manuscript. Please provide it for review. 

Author Response

Line 1: Change the entire document to MDPI Foods template (Check author guidelines: https://www.mdpi.com/journal/foods/instructions)

AU: Corresponding revisions have been made in the manuscript.

Line 20: The abstract should be a total of about 200 words maximum. Rewrite to reduce the word count.

AU: Corresponding revisions have been made in the manuscript. Please see the section “Abstract”.

Line 63: Needs further clarification on what exactly causes these changes.

AU: Corresponding revisions have been made in the manuscript. Please see line 45-46.

Line 55-65: The authors discuss well in this section, but there is no proper coherence, consequently, it should be rewritten. I ask authors to organize information. This version may make the reader confused. The final target should be clearly explained in the introduction. Could also explain why profiling the peptide composition of milk fermented by different strains are essential.

AU: As suggested by the reviewer, corresponding revisions have been made in the manuscript. Please see line 44-59.

Line 66-73: Very vague, can be rewritten to effectively give more context, include all latest publications.

AU: As suggested by the reviewer, corresponding revisions have been made in the revised manuscript. Please see line 44-59.

Line 77: Please clearly present the hypothesis and contribution of your study in the Introduction section.

AU: As suggested by the reviewer, corresponding revisions have been made in the manuscript. Please see line 60-67.

Line 93: Briefly explain what the modifications were.

AU: In our study, the eluted peptides were directly ionized in a mass spectrometer (Thermo Scientific, MA, USA) at an ion source voltage of 2.3 kV and subjected to a fully continuous MS scan followed by MS/MS scans.

Line 85: Was any other time considered other than 48 hrs.

AU: Yes. We analyzed the samples at 12h, 24h and 48h. We chose to show the samples at 48 h in the present study for those are the most representative.

Line 154: Experimental design is not clear. Was it a randomized approach to include all necessary experimental approaches in the text? Was it done in replications?

AU: As suggested by the reviewer, corresponding revisions have been made in the manuscript with detailed experimental design. Please see line 135.

Line 172-176: Further discussion is required

AU: Corresponding revisions have been made in the manuscript. Please see line 177-191.

Line 215: Further discussion is required

AU: Corresponding revisions have been made in the manuscript. Please see line 246-265.

Line 184: interesting; please provide further discussion

AU: Corresponding revisions have been made in the manuscript. Please see line 232-241.

Line 193: what does that indicate

AU: Herethat” indicated L. helveticus strains with different CEPs showed different cleave regions of αS1-casein hydrolysis. More specifically, DQHXN-Q32M42 (prtH2–3–4) could hydrolyze the most regions, followed by ATCC15009 (prtH3), and DYNDL36-6 (prtH2–4) could hydrolyze the least regions.

Line 219: Typically, in MDPI Foods Results and Discussion are written together; consider combining 

AU: As suggested by the reviewer, corresponding revisions have been made in the manuscript. Please see the section “Results and Discussion”.

Line 273: Being very wishful, please provide further clarifications from an industry standpoint

AU: We agree with the reviewer and rewrote the sentence. Please see line 270-275.

Line 287: Include a comment indicating the potential use of the proposed use at the industrial level.

AU: As suggested by the reviewer, corresponding revisions have been made in the manuscript. Please see line 268-270.

*The figures and tables are missing in the manuscript. Please provide it for review.

AU: The figures and tables are included in the manuscript.

Reviewer 2 Report

General comments:

The present manuscript is a well written study on different proteolytic activity of L.helveticus in milk. The authors present interesting peptidomic results, but the presentation of the results, especially the clarity of the figures should be improved. The authors define protein regions to categorize hydrolysis and preference in contrast to cleavage sites. This is an interesting approach, but are not compared or evaluated in the discussion part. The reason for selecting the different strains and the origin of the genomic data should be explained. 

Comments:

 Line 28: How are the cleavage regions defined? The protein sequence for the regions needs to be stated.

Line 100: Why was a 10 kDa membrane used? This will retain a considerable portion of larger peptides and should be discussed along with the length distribution of the peptides.

Line 117: Where genetic variants of caseins included in the database? Was the signal peptide included in the amino acid sequence from the database? Please specify.

Line 181: How many of the 65 peptides could be attributed to caseins? It would be good to have a percentage of peptides assigned to caseins or the bos taurus database in comparison to the total number of identified peptides.

Line 187: Figure 2a: Label on y axis in size of each list is missing. Figure description is not sufficient. Venn plot needs to be explained.

Line 194: It would be very helpful to have the table with the casein sections included in the paper. The partitioning of the caseins into different regions needs to be explained better. It should be specified if the regions were based on the results in Figure 3.

Line 202: Description of Figure 3 is unclear and not self-explanatory. What are the intensities in the figure? Meaning of peptides 1 22 for as1-casein and peptides 1 37 for b-casein are not explained. Protein sequence should be mentioned and explained. The plot above the heat map is not mentioned in the description.  

Line 212: How where the genes in the strains identified? There is not section about this in the material and no citation in this paragraph. Please add the missing information.

Line 220: Is it necessary to show the entire table? The majority of the peptidases are only active when cell lysis occurs and are not discussed further.

Line 249: As mentioned above, the total number of peptides and peptides assigned to milk proteins should be discussed. What was the percentage? Are there explanations for the distribution?

Line 256: In addition to the secondary structure of the proteins, the overall protein structure in the matrix should be considered. Caseins organized in micelles can result in different hydrolysis patterns compared to isolated caseins. The processing of the milk can result in protein modifications that affect hydrolysis.

Line 264: The authors mention in the introduction that the cleavage sites of L.helveticus strains are well described. It is however not discussed how the cleavage sites relate to the regions proposed by the authors.

Lines 272-278: How are the peptide profiles caused by different CEP profiles related to the specificity of the proteinases?  

Author Response

Line 28: How are the cleavage regions defined? The protein sequence for the regions needs to be stated.

AU: The cleavage regions were defined to showcase the common regions and the unique regions of various caseins among the three fermented samples. Taking αS1-casein as an example, region Ⅰ, Ⅳ, Ⅵ and Ⅷ are the common regions of the samples, while region Ⅱ and Ⅴ are the common regions of ATCC15009 and DQHXN-Q32M42, region Ⅶ is the common region of ATCC15009 and DYNDL36-6, and region Ⅲ is the unique region of DQHXN-Q32M42. Table S1 provides details of these regions.

Line 100: Why was a 10 kDa membrane used? This will retain a considerable portion of larger peptides and should be discussed along with the length distribution of the peptides.

AU: According to previous research by Maestri et al, the length of bioactive peptides in foods of animal origin varies from 2 to 40 amino acids with the molecular weight lower than 5 kDa. Therefore the 10 kDa membrane was chosen in the present study to cover the whole ranges of bioactive peptides.

Ref: Maestri, E.; Pavlievi, M.; Montorosi, M.; Marmiroli, N. Meta-Analysis for correlating structure of bioactive peptides in foods of animal origin with regard to effect and stability. Compr. Rev. Food Sci. Food Saf. 2019, 18, 3-30.

Line 117: Where genetic variants of caseins included in the database? Was the signal peptide included in the amino acid sequence from the database? Please specify.

AU: Relevant information of the Uniprot database has been added to the manuscript. The signal peptide was excluded from the amino acid sequence of the protein. Please see line 108-109.

Line 181: How many of the 65 peptides could be attributed to caseins? It would be good to have a percentage of peptides assigned to caseins or the bos taurus database in comparison to the total number of identified peptides.

AU: Of the 65 common peptides, 52 came from casein and the remaining 13 came from other proteins. As suggested by the reviewer, we have added the percentage of peptides assigned to caseins in comparison to the total number of identified peptides in the manuscript. Please see line 184-186.

Line 187: Figure 2a: Label on y axis in size of each list is missing. Figure description is not sufficient. Venn plot needs to be explained.

AU: As suggested by the reviewer, corresponding revisions have been made in the manuscript. Please see line 192, 186-187 and 183-195.

Line 194: It would be very helpful to have the table with the casein sections included in the paper. The partitioning of the caseins into different regions needs to be explained better. It should be specified if the regions were based on the results in Figure 3.

AU: As suggested by the reviewer, corresponding revisions have been made in the manuscript. Please see Table S1 for specific details of the regions.

Line 202: Description of Figure 3 is unclear and not self-explanatory. What are the intensities in the figure? Meaning of peptides 1 22 for as1-casein and peptides 1 37 for b-casein are not explained. Protein sequence should be mentioned and explained. The plot above the heat map is not mentioned in the description. 

AU: As suggested by the reviewer, corresponding revisions have been made in Figure 3. Each sample is represented on a separate line. For each residue (on the x axis), a green bar is drawn if this position is covered by at least one peptide in the given sample. The height of this bar is proportional to the count of peptides overlapping this position. The color intensity is proportional to the summed peak areas of peptides overlapping this position, with dark green indicating high peptide intensity and light green indicating low peptide intensity. The count of peptides overlapping a residue and the sum of their intensities is displayed by hovering over the bar. Please see line 218-219.

Protein sequences are available on “https://www.uniprot.org/”. All the protein sequences used in this manuscript have excluded the signal peptide region.

We sincerely thank the reviewer for careful reading, but, after careful consideration, we have decided to delete the Venn plot in this constitution diagram, which had no impact on the discussion of the article.

Line 212: How where the genes in the strains identified? There is not section about this in the material and no citation in this paragraph. Please add the missing information.

AU: The methods of CEP gene identification has been added to the material, please see the section “Analysis of CEP Genes” from line 75 to line 79.

Line 220: Is it necessary to show the entire table? The majority of the peptidases are only active when cell lysis occurs and are not discussed further.

AU: As suggested by the reviewer, the table has been revised. Please see line 160.

Line 249: As mentioned above, the total number of peptides and peptides assigned to milk proteins should be discussed. What was the percentage? Are there explanations for the distribution?

AU: As suggested by the reviewer, corresponding revisions have been made in the manuscript. Please see line 179-184.

Line 256: In addition to the secondary structure of the proteins, the overall protein structure in the matrix should be considered. Caseins organized in micelles can result in different hydrolysis patterns compared to isolated caseins. The processing of the milk can result in protein modifications that affect hydrolysis.

AU: As suggested by the reviewer, corresponding revisions have been made in the manuscript. Please see line 207-208.

Line 264: The authors mention in the introduction that the cleavage sites of L.helveticus strains are well described. It is however not discussed how the cleavage sites relate to the regions proposed by the authors.

AU: In the present study we used different approach and focused on the peptide profile and region preference of L. helveticus with different CEP genotypes. We are conducting analysis with more L. helveticus with different CEP genotypes to correlate the pattern of cleavage sites and peptide profile right now, and will provide more information in the on-going study.

Lines 272-278: How are the peptide profiles caused by different CEP profiles related to the specificity of the proteinases? 

AU: The αS1- and β-casein hydrolysis patterns were correlated with the CEP genotypes of L. helveticus strains, which were reflected in the extensiveness of their hydrolysis region of αS1-casein and the degree of β-casein hydrolysis. More specifically, DQHXN-Q32M42, the L. helveticus strain with prtH2–3–4, can cleave most regions of αS1-casein, followed by the strain with only prtH3, and the strain with prtH2–4 cleaves the least regions. Regarding β-casein, all three L. helveticus strains could cleave the whole region along the sequence of β-casein, except for the N-terminal region (signal peptide excluded). However, DQHXN-Q32M42, the strain with prtH2–3–4 generated more numerous, shorter in length and more widely distributed peptides than ATCC15009 (prtH3) and DYNDL36-6 (prtH2–4), and subsequently more prone to generate smaller bioactive peptides that possess biological activity.

Round 2

Reviewer 1 Report

I don't require additional clarifications. Thank you. 

Reviewer 2 Report

Dear Gao et al.

The comments and suggestions were taken into account and have improved the quality of the manuscript. I recommend the manuscript to be published in it´s current form.